# FairDen: Fair Density-Based Clustering

**Lena Krieger**[*,1], **Anna Beer**[*,2], **Pernille Matthews**[3], **Anneka Myrup Thiesson**[3], **& Ira Assent**[1,3]

[1] IAS-8: Data Analytics and Machine Learning, Forschungszentrum Jülich, Jülich, Germany
[2] Faculty of Computer Science, University of Vienna, Vienna, Austria
[3] Department of Computer Science, Aarhus University, Aarhus, Denmark

## ABSTRACT

Fairness in data mining tasks like clustering has recently become an increasingly important aspect. However, few clustering algorithms exist that focus on fair groupings of data with sensitive attributes. Including fairness in the clustering objective is especially hard for density-based clustering, as it does not directly optimize a closed form objective like centroid-based or spectral methods.

This paper introduces FairDen, the first fair, density-based clustering algorithm. We capture the dataset's density-connectivity structure in a similarity matrix that we manipulate to encourage a balanced clustering. In contrast to state-of-the-art, FairDen inherently handles categorical attributes, noise, and data with several sensitive attributes or groups. We show that FairDen finds meaningful and fair clusters in extensive experiments.

## 1 INTRODUCTION

Applying machine learning in critical situations, such as medical diagnostics or recidivism, requires fair and reliable models. It is crucial to ensure that minorities are not disadvantaged in the decision-making process based on machine learning models. This paper focuses on group-level fairness, i.e., treating different groups equally (Chhabra et al., 2021). These groups usually arise from sensitive attributes such as gender, race, or age.

Clustering is the task of grouping data objects based on similarity and is widely used in various domains. Recently, issues with fairness in data-driven models, particularly, machine learning, have been discussed, e.g., in connection with new legislation and laws regarding disparate impact (Dressel & Farid, 2018). Without adequately addressing sensitive attributes, most state-of-the-art clustering algorithms may produce *unfair* clusters: minorities might be disproportionately assigned to the same cluster, and people may suffer from biases in downstream tasks. For clustering, group-level fairness means avoiding favoring or discriminating against any sensitive group within each cluster (Chhabra et al., 2021), e.g., by making sure that the distribution of sensitive groups within each cluster is as close as possible to their overall distribution in the dataset (Chierichetti et al., 2017). The toy dataset in Figure 1 shows three spatial groups with a sensitive attribute denoted by the shape of the objects. A fair clustering consists of clusters with as many circles as triangles, corresponding to their overall distribution in the dataset. Clustering methods like $k$-Means (Lloyd, 1982) or $k$-Center (Har-Peled, 2008) can be adapted to a fair version (Bera et al., 2019; Bercea et al., 2018). However, incorporating fairness into density-based clustering is difficult, as this clustering notion does not directly optimize a closed form objective function like centroid-based or spectral methods.

Especially for real-world applications, fairness in density-based clustering methods is crucial as centroid-based methods have several shortcomings: 1) In contrast to density-based methods, centroid-based methods typically cannot handle categorical data, rendering them suboptimal for data with the most common sensitive attributes like 'gender', 'race', or 'marital status': these attributes usually have categorical values. 2) Density-based methods inherently manage noise, which is common in real-world applications, whereas partitioning-based clusterings may be more susceptible to the impact of noise. 3) For geographic data, travel distance along streets or railroads is often more expressive than the Euclidean distance, making density-connected clusters an inherently better fit. For example, an important task for city planning is to find school districts where the socio-economic

---

[*]Both authors contributed equally to this work and are ordered randomly.

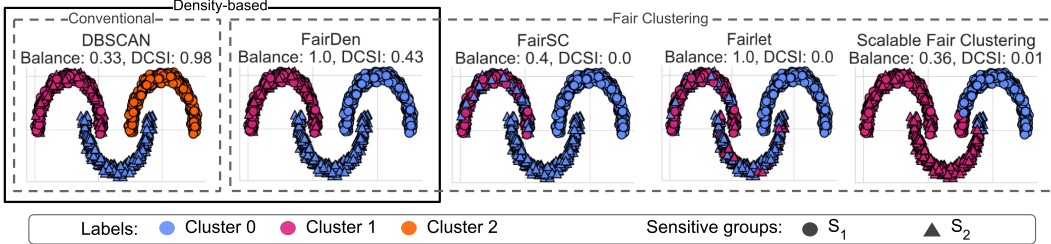

Figure 1: Different methods, balance, and clustering quality measured by DCSI (higher is better). A balanced density-based clustering has for each cluster the same ratio of circles to triangles as in the overall dataset, yielding a balance value of 1.0. Shapes indicate membership to one of two sensitive groups: the first moon has 50% triangles and 50% circles, the second moon has only triangles, the third moon has only circles, and all moons have the same number of data points. In contrast to our competitors, FairDen achieves a perfect balance and well-separated density-based clusters, indicated with a higher DCSI value. Colors indicate the two clusters found by FairDen and its competitors FairSC, clustering with Fairlets, Scalable Fair Clustering, respectively, the three density-based clusters found by DBSCAN.

distribution of schools is similar to ensure a fair education for all pupils. Density-based clusters offer good routes for school buses picking up the children and ensuring socio-economic fairness. However, there is no method yet that finds fair density-connected clusters.

In this work, we propose FairDen, a novel method that integrates fairness into density-based clustering. For this, we first capture the data's density-based structure, which may include categorical attributes. Capturing the structure allows us to flexibly manipulate the resulting similarity matrix based on fairness constraints. Our code is available at GitHub[1].

Our main contributions are as follows:

- We introduce FairDen, the first fair density-based clustering algorithm which successfully detects fair clusters of arbitrary shapes while enforcing a balance with respect to all sensitive attributes.
- FairDen is the first fair clustering algorithm that can handle mixed-type data and multiple sensitive attributes at once. Furthermore, it automatically detects noise points.
- Our experiments show that FairDen determines more balanced clusterings with respect to sensitive attributes than other density-based methods and detects density-based clusters better than other fair methods.

## 2 FAIRDEN: FAIR DENSITY-BASED CLUSTERING

In the following, we describe our novel density-based fair clustering method in detail. Including a fairness constraint into density-based clustering is challenging due to the characteristics of density-connectivity. At the heart of density-based clustering (e.g., DBSCAN (Ester et al., 1996)), it is defined as a binary property for each pair of points: Given a data set $\mathcal{X} = (x_1, \ldots, x_n)$ comprised of $n$ points, two points are density-connected if and only if there is a chain of *core* points connecting them. Core points lie in dense areas and are defined as points with at least $minPts$ points in their $\varepsilon$-radius for a given $minPts \in \mathbb{N}_{>1}$, $\varepsilon \in \mathbb{R}_{>0}$. Points form a chain if the (usually Euclidean) distance $d(p, q)$ between any two consecutive points is not larger than $\varepsilon$. A cluster is then a maximal set of density-connected points. This definition does not allow for an easy adaption of clusters to fulfill fairness criteria like balance between groups (Chierichetti et al., 2017; Bera et al., 2019) (cf. Section 2.2). Group-level fairness aims to achieve a ratio between sensitive groups within each cluster that corresponds to their ratio in the full dataset. However, the two most prominent existing approaches for including group-level fairness into clustering are not applicable to density-based clustering: Density-based methods usually do not have a differentiable clustering objective, thus, approaches like Bera et al. (2019) or Bercea et al. (2018) are not applicable. Other approaches use fairlets, i.e., minimal sets satisfying fairness that approximately preserve the $k$-clustering objective.

---

[1]https://jugit.fz-juelich.de/ias-8/fairden

However, those may destroy the density-connectivity structure of the dataset when locally merging points, as shown in Figure 1. Thus, to achieve fair density-connectivity-based clustering, we approach this problem from a novel perspective: we capture the density-connectivity between points in a continuous representation that admits group balancing, thereby presenting the first fair density-based clustering method. We describe how we capture the density-connectivity in Section 2.1, introduce fairness constraints in Section 2.2, and how to combine both in Section 2.3. In Section 2.4 we explain clustering with categorical data. A complexity analysis, limitations, and interpretation are in Section 2.5. The pseudo-code is in Appendix A.1, and our implementation is available online.

## 2.1 TRANSFORMING THE DENSITY-BASED OBJECTIVE

As outlined above, the major challenge in fair density-based clustering lies in the discrete nature of density-connectivity. Our idea is thus to transform the original problem into a continuous one that admits fairness balancing. Concretely, we capture the density-connectivity constraint by leveraging the density-connectivity distance (dc-distance) (Beer et al., 2023), which intuitively gives the smallest $\varepsilon$ such that two points are density-connected. While the density-connectivity between two points $p, q \in \mathcal{X}$ is still a discrete property, the dc-distance gives a mapping from the continuous space of possible $\varepsilon$-values onto the binary property by checking $d_{dc}(p, q) \leq \varepsilon$. The dc-distance represents pairwise distances between $n$ objects in a tree structure with objects as leaves and distances in the nodes. The hierarchy defined by the dc-distance works analogously to the well-known clustering hierarchy defined by the single-link distance in agglomerative single-linkage clustering. While usually the single link distance is based on the Euclidean distance, the dc-distance builds upon the mutual reachability distance used by DBSCAN, cf. Appendix B.2. Consequently, any partitioning based on the hierarchy defined by the dc-distance will give density-connected clusters. In FairDen, we observe that by casting the density-based clustering problem as one of the partitionings, optimizing a cut criterion on this hierarchy admits using spectral clustering methods (Beer et al., 2023). For this, we regard the following affinity matrix and its Laplacian (line 3 of Alg. 1):

$$\mathcal{A}_{ij} = 1 - \frac{d_{dc}(i, j)}{\max_{i,j} d_{dc}(i, j)} \tag{1}$$

$$\mathcal{L} = \mathcal{D} - \mathcal{A} \tag{2}$$

where $\mathcal{D}$ is the degree matrix. Minimizing the minCut as described in Beer et al. (2023) with ultrametric spectral clustering yields DBSCAN-like clusters. However, as we want to capture density-based clusters of potentially different densities, we cut the hierarchy given by the ultrametric at different levels instead of thresholding at a specific value for $\varepsilon$. This leads us to more prevalent versions of spectral clustering that detect the normalized cut (Hess et al., 2019) instead of the minCut. Thus, we apply $k$-means on the spectral embedding given by the first $k$ eigenvectors of our Laplacian $\mathcal{L}$ (see Von Luxburg (2007)) that we will adopt as shown in the following. Note that any clusters FairDen finds in the hierarchy given by the dc-distance follow the density-connectivity notion.

To find the normalized cut we solve for

$$\min_{\mathcal{H} \in \mathbb{R}^{n \times k}} Tr(\mathcal{H}^\top \mathcal{L} \mathcal{H}) \tag{3}$$

Where $\mathcal{H}$ encodes our clustering:

$$\mathcal{H}_{pl} = \begin{cases} 1/\sqrt{vol(C_l)} & p \in C_l \\ 0, & p \notin C_l \end{cases} \tag{4}$$

And $vol(C_l)$ is the volume of the cluster $C_l$, i.e., $vol(C_l) = \sum_{p \in C_l, q \in \mathcal{X}} \mathcal{A}_{pq}$. Similar to Kleindessner et al. (2019b), and as common in spectral clustering, we relax Eq. 4 to only requiring the following equality where $I_k \in \mathbb{R}^{k \times k}$ is the $k \times k$ identity matrix:

$$\mathcal{H}^\top D \mathcal{H} = I_k \tag{5}$$

Adopting the dc-distance as the basis for our similarity matrix in FairDen, we can now incorporate the fairness constraint in the form of a balancing matrix for density-based clustering, similar to the approach taken for spectral clustering in Kleindessner et al. (2019a).

## 2.2 FAIRNESS CONSTRAINT

As in recent work on fair clustering, our goal is to balance clusters regarding membership in sensitive groups (Kleindessner et al., 2019b; Chierichetti et al., 2017), where for each cluster, the ratio between different groups of specific sensitive attributes is as similar as possible to the ratio in the entire dataset. Where, e.g., FairSC (Kleindessner et al., 2019b) only considers individual sensitive attributes, FairDen handles also combinations of several sensitive attributes (e.g., *gender* and *race* simultaneously). Known as intersectional fairness (Gohar & Cheng, 2023), it has been shown that the combination of sensitive values (e.g., black woman) can lead to unique discrimination issues.

Our multi-sensitive notion takes into account membership in each sensitive group: Assume that each data object $x \in \mathcal{X}$ belongs to precisely one of $g_i$ many *sensitive groups* $S_{ij}$ of size $|S_{ij}| \geq k$ with $j = 0, ..., g_i - 1$ for each of its sensitive attributes $S_i$, with $i = 0, ..., a - 1$ and $k$ being the number of clusters: $\forall S_i : \mathcal{X} = \dot{\bigcup}_j S_{ij}$

We want to maximize the group-level balance (Chierichetti et al., 2017; Bera et al., 2019) that assesses the ratio $r_{ij}(c)$ of all sensitive groups $S_{ij}$ within each cluster $c$ to $r_{ij}$, the distribution of the sensitive groups within the whole dataset.

We define $r_{ij} = \frac{|S_{ij}|}{n}$ and $r_{ij}(C_l) = \frac{|S_{ij} \cap C_l|}{|C_l|}$, where $|S_{ij} \cap C_l|$ is the number of samples belonging to group $S_{ij}$ within a cluster $C_l$. The balance for a cluster $C_l$ (see Eq. 6) is the minimum balance value for each sensitive group $S_{ij}$ within the cluster, see Eq. 7.

$$\text{balance}(C_l) = \min_{\forall S_{ij}} \text{balance}_{ij}(C_l) \tag{6}$$

$$\text{balance}_{ij}(C_l) = \min\left(\frac{r_{ij}}{r_{ij}(C_l)}, \frac{r_{ij}(C_l)}{r_{ij}}\right) \tag{7}$$

To handle several sensitive attributes simultaneously, we introduce combined group membership vector $f_p^{S_x}$, where $S_x$ denotes membership in some combination of sensitive groups across sensitive attributes. Thus, instead of $g_i$ groups for every of the $a$ attributes, we consider up to $\sum_{i=1}^{a} g_i$ combined sensitive groups $S_x$ (e.g., *Asian + female*, *Asian + male*). Due to this representation, FairDen has the advantage of different sensitive attributes combined being weighed equally, and attributes with fewer groups (e.g., *gender*) do not impact the clustering more than attributes with more groups (e.g., *race*), as would be the case for simple concatenation of sensitive attributes.

For a balanced clustering, the ratio of each combined sensitive group $S_x$ within each cluster $C_l$ should be as similar as possible to its ratio in the overall dataset:

$$\forall C_l, S_x : \frac{|S_x \cap C_l|}{|C_l|} \approx \frac{|S_x|}{n} \tag{8}$$

Thus, our FairDen objective for a perfect balance is

$$\forall C_l, S_x : \frac{|S_x \cap C_l|}{|C_l|} - \frac{|S_x|}{n} = 0 \tag{9}$$

With that and Eq. 4, we reformulate Eq. 9

$$\forall C_l, S_x : \sum_{p \in \mathcal{X}} \left(f_p^{S_x} - \frac{|S_x|}{n}\right) \mathcal{H}_{pl} = 0 \tag{10}$$

To solve this equation, we use fairness matrix $\mathcal{F} \in \mathbb{R}^{n \times |S_x|}$ with vectors $f_p^{S_x} - (|S_x|/n) \cdot \mathbf{1}_n$ as columns for the first $|S_x| - 1$ combined sensitive groups[2] (Alg. 1 line 4). The fairness matrix captures the term in brackets in Eq. 10, thereby balancing the clustering regarding the sensitive groups. Then, we include Eq. 10 as a constraint into our overall clustering objective from Eq. 3:

$$\mathcal{F}^\top \mathcal{H} = 0 \tag{11}$$

By enforcing this constraint we make sure that any cluster assignment $\mathcal{H}$ yields a balanced clustering with respect to any sensitive group as mandated by the fairness matrix $\mathcal{F}$.

---

[2] we omit a membership vector for the last combined sensitive group s.t. the matrix is not over-determined

## 2.3 INTEGRATING DENSITY-CONNECTIVITY AND FAIRNESS

FairDen integrates the density-connectivity objective (Eq. 3, 5) with the fairness constraint (Eq. 11):

$$\min_{\mathcal{H}\in\mathbb{R}^{n\times k}} Tr(\mathcal{H}^\top \mathcal{L}\mathcal{H}) \text{ subject to } \mathcal{H}^\top \mathcal{D}\mathcal{H} = \mathcal{I}_k \text{ and } \mathcal{F}^\top \mathcal{H} = 0 \tag{12}$$

While this objective bears superficial similarity with the one for spectral clustering in Kleindessner et al. (2019b), there are fundamental differences in the two approaches: Most importantly, our Laplacian $\mathcal{L}$ in FairDen captures density-connectivity and is unrelated to the Laplacian of Kleindessner et al. (2019b). Furthermore, our fairness constraints handle multiple sensitive attributes. We now reshape Eq. 12 so that it can be solved by spectral clustering, i.e., by eigendecomposition and subsequent application of $k$-means on the first $k$ eigenvectors. For this, we ensure the condition in Eq. 11 is met (Alg. 1 line 5): we compute the orthonormal basis of the nullspace of $\mathcal{F}^\top$, as the columns of a matrix $\mathcal{Z} \in \mathbb{R}^{n \times (|S_x|-1)}$, so that we can replace $\mathcal{H} = \mathcal{Z}\mathcal{Y}$ for a $\mathcal{Y} \in \mathbb{R}^{(n-|S_x|+1)\times k}$:

$$\min_{\mathcal{Y}} Tr(\mathcal{Y}^\top \mathcal{Z}^\top \mathcal{L}\mathcal{Z}\mathcal{Y}) \text{ subject to } \mathcal{Y}^\top \mathcal{Z}^\top \mathcal{D}\mathcal{Z}\mathcal{Y} = \mathcal{I}_k \tag{13}$$

We want to express the condition in the shape $\mathcal{V}^\top \mathcal{V} = \mathcal{I}_k$. We make use of the fact that $\mathcal{Z}^\top \mathcal{D}\mathcal{Z}$ is positive definite and square (as it comes from a fully connected graph given by dc-distances that are calculated for *all* pairs): This means, there is a $\mathcal{Q} \in \mathbb{R}^{(n-|S_x|+1)\times(n-|S_x|+1)}$ so that: $\mathcal{Q}^2 = \mathcal{Z}^\top \mathcal{D}\mathcal{Z}$ (Alg. 1 lines 6 and 7). With this, we can replace $\mathcal{Y} = \mathcal{Q}^{-1}\mathcal{V}$ for some $\mathcal{V} \in \mathbb{R}^{(n-|S_x|+1)\times k}$ in Eq. 13:

$$\min_{\mathcal{V}} Tr(\mathcal{V}^\top \mathcal{Q}^{-1^\top} \mathcal{Z}^\top \mathcal{L}\mathcal{Z}\mathcal{Q}^{-1}\mathcal{V}) \text{ subject to } \mathcal{V}^\top \mathcal{V} = \mathcal{I}_k \tag{14}$$

Now our solution $\mathcal{V}$ to Eq. 14 can be found by computing the eigenvalue decomposition of $\mathcal{Q}^{-1^\top}\mathcal{Z}^\top \mathcal{L}\mathcal{Z}\mathcal{Q}^{-1}$ and the $k$ eigenvectors belonging to the $k$ smallest eigenvalues become the columns of $\mathcal{V}$ (Alg. 1 line 8). We obtain a fair density-based clustering by applying $k$-means on $\mathcal{H} = \mathcal{Z}\mathcal{Q}^{-1}\mathcal{V}$ (Alg. 1 line 9). Note that subdividing the hierarchy given by the dc-distance automatically detects noise points as subtrees that are smaller than the given $\mu$. Our algorithm thus returns a $k$-means clustering that gives us $k$ non-noise clusters (Alg. 1 lines 10-18).

## 2.4 CATEGORICAL ATTRIBUTES AND MULTIPLE SENSITIVE ATTRIBUTES

While most fair clustering methods (cf. Section 3.4) target numerical data or assume a given similarity matrix (Kleindessner et al., 2019b), this is in strong contrast to their real-world use cases: Most data for which fairness is fundamental contains categorical attributes, especially considering sensitive attributes are usually categorical. Thus, handling categorical data, while preserving the underlying information, in fair clustering is of utmost importance.

In FairDen, we provide full inclusion of categorical attributes and mixed-type data. For this, we compute a similarity matrix $S_G$ based on Goodall1 (Boriah et al., 2008). We extend Eq. 1 to a weighted average of dc-distance and categorical similarity measure. Consider data set $\mathcal{X} = (x_1, \ldots, x_n)$ with $n$ $d$-dimensional points where the $d$ dimensions consist of $d_n$ numerical and $d_c$ categorical features. Instead of using Eq. 1, we can then compute our affinity matrix as follows:

$$\mathcal{A}_{ij} = \frac{d_n}{d}\left(1 - \frac{d_{dc}(i,j)}{\max_{i,j} d_{dc}(i,j)}\right) + \frac{d_c}{d} \cdot S_G(i,j) \tag{15}$$

Like this, we combine both types of data weighted proportionally to the number of numerical and categorical features within the dataset. Note that this approach is not suitable for $k$-means-like algorithms, as they need a metric space where centroids can be computed: Clustering methods solving a $k$-objective would require an embedding of the categorical data first. In contrast, density-based methods only require a distance or similarity matrix. FairDen captures arbitrarily shaped clusters with the dc-distance. Since the dc-distance is defined on numerical data, the input data has to feature some numerical attributes. With the above extension, we allow for the inclusion of categorical attributes in a mixed data type approach.

When including multiple sensitive attributes, we combine those sensitive attributes into one meta-sensitive attribute, such that FairDen balances the combined groups. For example, for sensitive attributes *race* and *gender* the combinations might include groups like 'Female-Asian'. Clearly, multiple sensitive attributes are highly complex to handle, and achieving balance also requires a sufficient amount of data for each subgroup, for a fair clustering to be attainable.

## 2.5 ANALYSIS

**General Interpretation** FairDen aims for optimal balance (see Eq. 9 and following). By working on the hierarchy of density-based clusterings given by the dc-distance, all clusterings by spectral cuts are density-based. However, we relax DBSCAN's clustering criterion by allowing clusters of different densities, and by prioritizing balance. Thus, as Section 3.5 shows, FairDen finds clusterings with high balance values. Their density-connectivity is of course slightly lower than the non-fair density-connectivity of DBSCAN clusterings. However, FairDen captures arbitrarily-shaped clusters better than any other fair clustering algorithm, see e.g. Figure 1, leading to higher clustering quality regarding density-based clusters throughout our experiments in Section 3.5.

**Complexity Analysis** The complexity of FairDen in the number of data points is dominated by the eigenvector decomposition in $O(n^3)$. As we need only up to $k$ (number of clusters) eigenvectors, this step can be accelerated significantly by, e.g., using power iterations. After computing the dc-distance matrix, the runtime is independent of the dimensionality, yielding linear dependency in dimensionality $d$. Thus, the complexity is in $O(n^3 \cdot d)$, comparable to other fair clustering methods.

**Limitations** Note that a balanced clustering with $k$ clusters is only possible if every sensitive group has at least $k$ members. In extreme cases with insufficient data for particular subgroups, these individuals are not considered - here, a manual check regarding minorities with very few members is advisable. For several sensitive attributes, FairDen optimizes the balance of their combination, and not of the *individual* sensitive attributes, which may or may not be adequate depending on the application goal.

## 3 EXPERIMENTAL EVALUATION

We measure cluster quality with DCSI (Gauss et al., 2024) and group-level fairness with generalized balance Bera et al. (2019) (Sect. 3.2,3.1). We study real-world benchmarks (Sect. 3.3) and compare to state-of-the-art fair clustering methods FairSC (Kleindessner et al., 2019b), normalized FairSC (Kleindessner et al., 2019b), Fairlets (Chierichetti et al., 2017), and Scalable Fair Clustering (Backurs et al., 2019) (Section 3.4). Following Schubert et al. (2017), we fix the parameter $\mu = 2d - 1$ for FairDen and show an ablation in App. A.2. Further technical details in App. C.1.

## 3.1 EVALUATING CLUSTERING QUALITY

Note that ground truth labels of benchmark data are not typically balanced, but rather potentially include bias. Crucially, these labels cannot serve as external measures for fairness, but only for (biased) group assignments. Thus, an optimal correspondence to the ground truth clustering is not necessarily desirable. We aim to find a *fair*, balanced clustering, *rather than* replicating the biases in original labels. To evaluate the clustering quality, we compare standard clustering with DBSCAN to that of our fair results using normalized mutual information (NMI) (Danon et al., 2005) and adjusted rand index (ARI) (Hubert & Arabie, 1985) where higher values indicate better results. When assessing NMI and ARI regarding DBSCAN results (denoted as $\text{NMI}_{DB}$, $\text{ARI}_{DB}$), we exclude data points marked as noise by DBSCAN in both solutions when computing these measures. DBSCAN's hyperparameters are optimized as described in Appendix C.1. We additionally include DCSI (Gauss et al., 2024) as an internal evaluation measure to assess quality of density-based clusters of arbitrary shape. DCSI evaluates the separation and connectivity of a cluster based on the density-connectivity in DBSCAN (Ester et al., 1996) (details in App. C.2). We follow good evaluation practices as recommended in Ullmann et al. (2023). Thus we do not use other internal measures like Silhouette Coefficient (Rousseeuw, 1987) or Dunn Index as they are not suitable for clusters with non-convex shapes (Gauss et al., 2024). For clusterings $\mathcal{C}^\nu$ that include noise labels, we compute NMI, ARI, and DCSI values based on all non-noise labeled points and multiply the results with the percentage of non-noise points. This accounts for the extent of data that is actually assigned to clusters. Note that detecting any noise prevents an optimal score, introducing a bias *against* noise-detecting methods like FairDen. However, detecting noise is essential for real-world use cases where the percentage of noise is unknown beforehand.

Table 1: Properties of our group-level fair competitors.

| Algorithm | Fairlets | Scalable Fair Clustering | Fair SC | FairDEN(**ours**) |
|---|---|---|---|---|
| Density-based | ✗ | ✗ | ✗ | ✓ |
| Multiple sensitive attributes | ✗ | ✗ | ✗ | ✓ |
| Multiple ($> 2$) sensitive groups | ✗ | ✗ | ✓ | ✓ |
| Categorical features | ✗ | ✗ | ✗ | ✓ |

## 3.2 EVALUATING FAIRNESS

Building on Equations 6 and 7, we report the balance (Chierichetti et al., 2017; Bera et al., 2019) for a clustering $\mathcal{C}$ with $k$ clusters as the average balance over the individual clusters, with values between 0 and 1 where higher values imply fairer results. Note that, while balance is one of the most common measures in the area of fairness in clustering, it has some limitations to be considered during evaluation and analysis:

- If a sensitive group is smaller than the number of clusters, at least one cluster has a balance of 0, significantly reducing the overall balance.
- As the balances are computed per cluster, a higher balance does not necessarily mean that the majority of points are in fair clusters, as cluster sizes might vary heavily.
- Some existing works only report the balance of the largest clusters (e.g., Bera et al. (2019)), making a direct comparison difficult.

In order to ensure a fair evaluation in the presence of noise, we adjust the balance calculation by excluding noise-labeled points for the calculation and multiplying the result by the percentage of non-noise points, similar to the evaluation of clustering quality.

## 3.3 REAL WORLD DATA

We use the common benchmark datasets for fair clustering (Chhabra et al., 2021; Le Quy et al., 2022), details shown in Table 6: The datasets *Adult* (Kohavi et al., 1996), *Bank* (Moro et al., 2014), *Communities and Crime* (Asuncion & Newman, 2007), and *Diabetes* (Strack et al., 2014) provide different scenarios in terms of dimensionality and number of sensitive groups. Achieving a balanced clustering for sensitive groups with fewer members than the number of clusters ($k$) is not feasible (e.g., if a minority encloses only 2 members, they cannot be distributed across 5 clusters). Thus, we include all data objects belonging to sensitive groups of sufficient size: $|S_{ij}| \geq k$.

## 3.4 COMPETITORS

We compare FairDen with state-of-the-art fair clustering methods aiming at group-level fairness. Table 1 shows important properties of FairDen and its competitors: FairDen closes several important gaps, as most state-of-the-art fair clustering methods handle neither categorical features nor multiple sensitive attributes. We compare to the following methods that we discuss in more detail in Section 4:

**FairSC** and **normalized FairSC** (Kleindessner et al., 2019b) need a weighted adjacency matrix as input. Thus, as common in spectral clustering, we employ a kNN graph with $k = 15$ based on the numerical features and sensitive attributes to apply those methods also on tabular data. **Fairlet** and **Fairlet MCF** (Chierichetti et al., 2017) as well as **Scalable Fair Clustering** (Backurs et al., 2019) follow a two-stage approach based on identifying fairlets. Since the construction of fairlets is an $NP$-hard problem both researchers present an approximative method, which are included in our experiments. The fairlet approach limits the methods to one binary sensitive attribute. Thus, they cannot be used in all our experiments.

## 3.5 EXPERIMENTS

**Fair Clustering of Real-World Benchmark Data** Figure 2 shows that FairDen consistently reaches the highest or competitive balance values for the real-world datasets when regarding only one sensitive attribute. Note that Fairlet and Scalable Fair Clustering cannot be applied to datasets with non-binary sensitive attributes. Table 2 shows (along the balance values) also the evaluation of

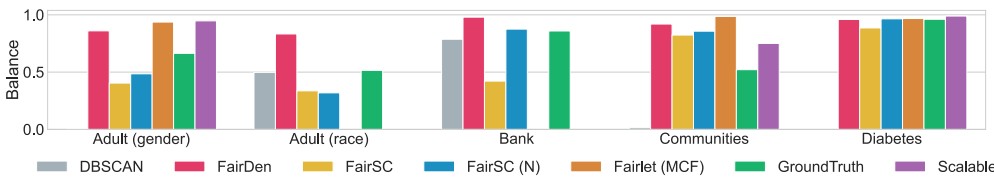

Figure 2: Balances for all competitors and benchmark datasets. Fairlet (MCF) and Scalable Fair Clustering are not included for settings including non-binary sensitive groups.

Table 2: Number of clusters $k$, Balance, DCSI, ARI for real-world benchmark data. Diabetes dataset for $k$=2 (Ground truth) and $k$=4 (DBSCAN clusters) respectively.

| | k | Algorithm | Balance | DCSI | ARI | | k | Algorithm | Balance | DCSI | ARI |
|---|---|---|---|---|---|---|---|---|---|---|---|
| **Adult (gender)** | 2 | DBSCAN | 0.01 | **0.97** | 0.00 | **Communities** | 2 | DBSCAN | 0.01 | **0.65** | -0.03 |
| | 2 | FairDen | 0.86 | 0.04 | 0.05 | | 2 | FairDen | 0.92 | 0.15 | 0.09 |
| | 2 | FairSC | 0.40 | 0.00 | 0.23 | | 2 | FairSC | 0.82 | 0.13 | 0.03 |
| | 2 | FairSC (N) | 0.49 | 0.00 | 0.27 | | 2 | FairSC (N) | 0.86 | 0.13 | 0.03 |
| | 2 | Fairlet (MCF) | 0.94 | 0.00 | 0.00 | | 2 | Fairlet (MCF) | **0.99** | 0.05 | 0.16 |
| | 2 | GroundTruth | 0.66 | 0.00 | 1.00 | | 2 | Scalable | 0.75 | 0.08 | -0.03 |
| | 2 | Scalable | **0.95** | 0.01 | -0.01 | | 2 | GroundTruth | 0.52 | 0.07 | 1.00 |
| **Adult (race)** | 2 | DBSCAN | 0.50 | **0.99** | 0.02 | **Diabetes** | 2 | DBSCAN | - | - | - |
| | 2 | FairDen | **0.83** | 0.09 | 0.05 | | 2 | FairDen | 0.96 | **0.08** | 0.01 |
| | 2 | FairSC | 0.34 | 0.00 | -0.03 | | 2 | FairSC | 0.89 | 0.00 | -0.01 |
| | 2 | FairSC (N) | 0.32 | 0.00 | 0.16 | | 2 | FairSC (N) | 0.96 | 0.00 | 0.01 |
| | 2 | Fairlet (MCF) | - | - | - | | 2 | Fairlet (MCF) | 0.97 | 0.00 | 0.00 |
| | 2 | Scalable | - | - | - | | 2 | Scalable | **0.99** | 0.01 | 0.00 |
| | 2 | GroundTruth | 0.52 | 0.00 | 1.00 | | 2 | GroundTruth | 0.96 | 0.00 | 1.00 |
| **Bank** | 2 | DBSCAN | 0.79 | **0.99** | 0.01 | | 4 | DBSCAN | 0.01 | **0.88** | - |
| | 2 | FairDen | **0.98** | 0.14 | 0.21 | | 4 | FairDen | 0.95 | 0.24 | - |
| | 2 | FairSC | 0.42 | 0.00 | -0.06 | | 4 | FairSC | 0.23 | 0.01 | - |
| | 2 | FairSC (N) | 0.88 | 0.00 | -0.04 | | 4 | FairSC (N) | 0.61 | 0.19 | - |
| | 2 | Fairlet (MCF) | - | - | - | | 4 | Fairlet (MCF) | 0.95 | 0.00 | - |
| | 2 | Scalable | - | - | - | | 4 | Scalable | **0.96** | 0.07 | - |
| | 2 | GroundTruth | 0.86 | 0.00 | 1.00 | | 4 | GroundTruth | - | - | - |

clusterings, i.e., DCSI and ARI. Density-connected structures become more significant for higher numbers of clusters, while most benchmark datasets have binary classification as ground truth. Consequently, the DCSI values are very low as soon as a few points are assigned differently than their density-connectivity suggests. Nevertheless, FairDen achieves the highest DCSI besides DBSCAN for almost all datasets while improving the balance by a large amount.

**Multiple sensitive attributes** While our competitors (Chierichetti et al., 2017; Backurs et al., 2019; Kleindessner et al., 2019b) cannot balance a clustering with respect to several sensitive attributes at once, FairDen can include an arbitrary number of sensitive attributes, successfully balancing the sensitive attributes as Figure 3 shows. We regard fairness with respect to all three sensitive attributes of the Adult dataset and study various combinations of sensitive attributes in FairDen. Each heat map shows the balance regarding the *individual* sensitive attributes. Note that this is different from balancing the combined groups. In the first heat map, each row represents the outcomes when labeling exactly one of the attributes as sensitive, while the remaining are included as non-sensitive categorical attributes. As expected, the balance for any sensitive attribute is highest in the row where it was included as sensitive in FairDen: each column's highest value lies on the diagonal. The second heat map shows results when including two sensitive attributes simultaneously (as indicated by the row description), yielding Pareto-optimal balance values for the respective attributes. The third heat map reveals an interesting effect: although all balance values are notably high (surpassing any combination for a single sensitive attribute), it seems like we could achieve a better-balanced cluster when using only *marital status* and *race* as sensitive attributes (row three of second heat map "M&R"). However, when inspecting the results and the underlying clustering, it is apparent that combining all three attributes results in more combinations of sensitive groups, consequently rendering the fairness constraint more complex than for just two sensitive attributes. When evaluating the balance across the combined sensitive groups, we can observe that including three sensitive attributes results in a higher balance (0.32) regarding the combination than "M&R" (0.25). Thus, the consideration of combined sensitive groups may not always yield the optimal solu-

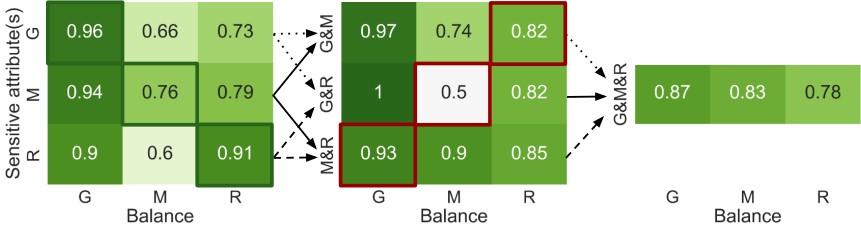

Figure 3: Columns show balance with respect to sensitive attributes *gender* (G), *marital status* (M) and *race* (R) in FairDen clusterings of the Adult dataset. Left: results for clustering when including only one of the sensitive attributes, *highest* column-wise values/balance for the attribute labeled as sensitive is framed in green. Middle: results for clusterings when including two of the sensitive attributes as indicated by the row description. *Lowest* column-wise values/ balance of attribute that is *not* labeled as sensitive is framed in red. Right: results for including all three sensitive attributes.

tion in terms of proportionally distributing individual sensitive attributes, but it does ensure the most balanced distribution across all combinations.

**Categorical data**  In contrast to our competitors, FairDen inherently includes categorical attributes. We conducted a comparison of results, excluding (FairDen-) and including (FairDen) categorical attributes in Table 3, with sensitive attributes as indicated. As the DCSI is not defined for data with categorical attributes, we only report the correspondence to DBSCAN results (columns $ARI_{DB}$ and $NMI_{DB}$). The results, see Table 3, show that both versions usually improve the balance with respect to the ground truth. When incorporating categorical attributes, the balance increases. The correspondence to original DBSCAN clusters stays approximately the same for both versions, i.e., the density-connectivity aspect is not negatively influenced by including categorical attributes. Lastly, noise levels consistently remained low for the three datasets.

Table 3: Comparison of results when excluding/including (FairDen-/ FairDen) categorical attributes for datasets Adult (sensitive attribute: gender/ race) and Bank. Note that none of our fair competitors is able to work with categorical attributes.

|  | Algorithm | Balance | $ARI_{DB}$ | $NMI_{DB}$ | Noise |
|---|---|---|---|---|---|
| Adult (g) | DBSCAN | 0.01 | 1.00 | 1.00 | 0.99 |
| | FairDen | **0.96** | **0.00** | **0.00** | **0.00** |
| | FairDen- | 0.86 | 0.00 | 0.00 | 0.00 |
| | Ground Truth | 0.66 | -0.04 | 0.01 | - |
| Adult (r) | DBSCAN | 0.50 | 1.00 | 1.00 | 0.00 |
| | FairDen | **0.86** | 0.01 | 0.01 | **0.00** |
| | FairDen- | 0.83 | 0.01 | 0.02 | 0.00 |
| | Ground Truth | 0.52 | 0.01 | 0.01 | - |
| Bank (m) | DBSCAN | 0.79 | 1.00 | 1.00 | 0.00 |
| | FairDen | **0.99** | 0.01 | 0.01 | **0.00** |
| | FairDen- | 0.98 | 0.01 | 0.01 | 0.00 |
| | Ground Truth | 0.86 | 0.00 | 0.00 | - |

**Robustness with respect to number of clusters**

In all previous experiments, the number of clusters ($k$) was determined based on the ground truth of the dataset or DBSCAN clustering (cf. Appendix C.1). In Figure 4, we assess the performance and balance of all algorithms under consideration for various numbers of clusters. Note that the Fairlet (MCF) and Scalable Fair Clustering approaches cannot handle data with non-binary sensitive attributes. As a result, for the sensitive attribute *race* (depicted on the left), comparisons can only be made with FairSC versions. In this context, FairDen constantly receives the highest balance (top left) while achieving comparable quality measured by DCSI. When *gender* is considered as the sensitive attribute (on the right), FairDen exhibits the highest DCSI up to $k = 4$, and is only surpassed by Scalable and the FairSC variants for a larger number of clusters. Notably, all methods except FairSC versions yield relatively high balance values.

## 4 RELATED WORK: GROUP-LEVEL FAIR CLUSTERING USING BALANCE

Several fair clustering methods have been developed to ensure group-level fair clustering. We refer to the notion of group-level fairness as introduced by Chierichetti et al. (2017), which aims to ensure that each group is represented in each cluster in proportions similar to the overall dataset. Other definitions such as proportional fairness (Chen et al., 2019) also exist. Many works have approached this problem from different angles, e.g., including probabilistically defined group-membership (Esmaeili et al., 2020), treating it as a constrained optimization problem (Esmaeili et al., 2021), and

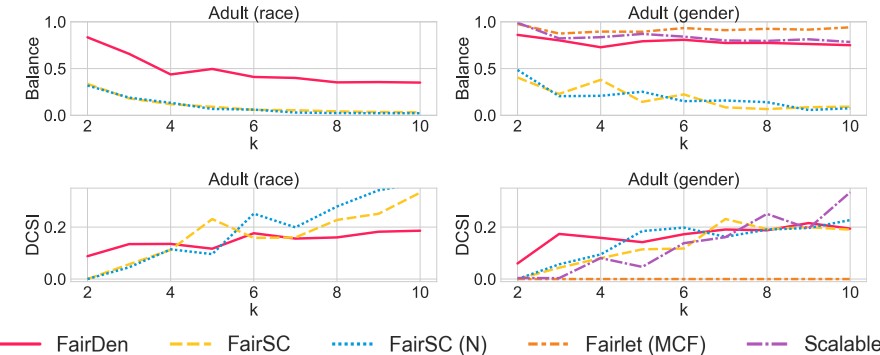

Figure 4: Balance (top) and DCSI (bottom) depending on number of clusters $k$. Gender has only two sensitive groups, race has five sensitive groups, which some competitors cannot handle.

integrating different notions of group-level fairness (Dickerson et al., 2024). Our focus is on work that optimizes balance.

Chierichetti et al. (2017) and Bercea et al. (2018) proposed two-stage approaches. Chierichetti et al. (2017) introduced the concept of *fairlets*, which are minimal sets of points with a balanced distribution. These fairlets are then used as input for a subsequent clustering method, achieving a balanced clustering that can approximately preserve clustering objectives for $k$-center or $k$-means. Bercea et al. (2018) solve a fair reassignment problem with a linear program (LP) based on weakly supervised rounding. However, these methods are more suitable for centroid-based approaches and are not appropriate for density-connected clusters: These minimal subsets can influence the density-connectivity between clusters significantly, leading to a potentially very poor approximation. This effect is evident in Figure 1, where clustering a three moons toy dataset with a binary sensitive attribute using fairlets in density-connected structures consisting of only one sensitive group (e.g., the middle cluster) leads to an undesirable patchwork of points in the left and middle cluster. Similar to Bercea et al. (2018), Bera et al. (2019) and Ahmadian et al. (2019) present LP-based algorithms. Bera et al. (2019) aims for a balance within bounds depending on a user-input $\alpha$ and $\beta$, which differs from the focus of this paper, where we optimize the balance objective. Based on the solution returned by a clustering algorithm, they solve a fair assignment problem with an LP. Note that, while this approach allows non-binary sensitive attributes, it is still only defined for $k$-center, $k$-median, and $k$-means objective. Ahmadian et al. (2019) solve the $k$-center objective with bicriteria approximation guarantee, bounding the additive violation. Scalable Fair Clustering (Backurs et al., 2019) partitions the dataset into fairlets that are merged into $k$ clusters. The merging alleviates problems caused by fairlets in similar approaches (see Figure 1) so that clusters are more contiguous. Fair Spectral Clustering (FairSC) (Kleindessner et al., 2019b) is a group-level fair spectral clustering algorithm that builds on an affinity graph. It comes in two versions: unnormalized FairSC, which is based on ratio cuts, and normalized FairSC, which is based on normalized cuts. However, FairSC can handle only one sensitive attribute and lacks details on how to obtain the affinity graph from tabular data.

Note that all these approaches do not consider density or density-connectivity, have limitations in handling noise, and do not accommodate categorical data in the clustering process. Additionally, methods based on fairlets struggle with multiple sensitive attributes or more than two sensitive groups. These limitations are summarized in Table 1.

## 5 CONCLUSION

We introduced FairDen, the first density-based fair clustering algorithm. By imposing fairness to the density-connectivity distance we transformed the problem to a graph cut solvable with spectral clustering. FairDen is, to the best of our knowledge, the first group-level fair clustering algorithm that incorporates categorical data, multiple sensitive attributes, and non-binary sensitive attributes at once. Extensive experiments show that FairDen indeed finds fair and density-based clusters. Future work includes investigating different fairness notions, handling micro-clusters, and exploiting GPU parallelism and Nyström approximation (Hohma et al., 2022) for runtime improvement.

ACKNOWLEDGEMENTS

This work was partially funded by project W2/W3-108 Initiative and Networking Fund of the Helmholtz Association and partially supported by the Pioneer Centre for AI, DNRF grant number P1.

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

# A  FAIRDEN

## A.1  PSEUDO-CODE

The pseudo-code for our novel fair density-based clustering method FairDen is given in Algorithm 1:

---

**Algorithm 1** FairDen

---

**Input:** $\mathcal{X} \in \mathbb{R}^{n \times d}$ dataset, $G \in \mathbb{R}^{n \times a}$ group-membership vectors for each of the $a$ sensitive attributes, $\mu, k$
**Output:** assignment of point in $\mathcal{X}$ to one of $k$ cluster indices or to index $-1$ for noise
 1: $D_{dc} \leftarrow$ dc-distance$(\mathcal{X})$ pairwise dc-distances in $\mathcal{X}$
 2: $\mathcal{A} \leftarrow 1 - D_{dc}/max(D_{dc})$ get affinity matrix (Eq. 1)
 3: $\mathcal{L} \leftarrow \mathcal{D} - \mathcal{A}$ get Laplacian (Eq. 2)
 4: $f_p^{S_x} \leftarrow$ combined_membership$(G)$ membership vector to combined sensitive group $S_x$ for each point $p \in \mathcal{X}$
 5: $\mathcal{F} \leftarrow f_p^{S_x} - \frac{|S_x|}{n} * \mathbf{1}_n$ get Fairness matrix
 6: $\mathcal{Z} \leftarrow$ orth_basis(nullspace$(\mathcal{F}^{\top})$) get orthonormal basis for the nullspace of $\mathcal{F}^{\top}$ (Eq. 11)
 7: $\mathcal{Q} \leftarrow \sqrt{\mathcal{Z}^{\top}\mathcal{D}\mathcal{Z}}$
 8: $\mathcal{V} \leftarrow k$ smallest eigenvectors of $(\mathcal{Q}^{-1})^{\top}\mathcal{Z}^{\top}\mathcal{L}\mathcal{Z}\mathcal{Q}^{-1}$
 9: $\mathcal{H} \leftarrow \mathcal{Z}\mathcal{Q}^{-1}\mathcal{V}$
10: Clustering $\mathcal{C} \leftarrow k$-means$(\mathcal{H})$ compute clustering
11: **while** $|C_l| < \mu$ (for any cluster $C_l$) **do**
12:     $C_N \leftarrow C_l$, for $|C_l| < \mu$
13:     **if** $|\mathcal{C} \setminus C_N| < k$ **then**
14:         repeat step 10 for $k := k + 1$
15:     **else**
16:         **return** $\mathcal{C}$
17:     **end if**
18: **end while**

---

## A.2 MIN POINTS ABLATION

Table 4 gives a small ablation study for the hyperparameter $\mu$ that we set, according to Schubert et al. (2017), to $\mu = 2d - 1$ for all experiments in the main part of the paper. According rows are marked in bold. While there is no free lunch, the heuristic yields a good trade-off between clustering quality measured by DCSI and balance for all data sets.

Table 4: Overview of DCSI, balance and noise (in %) results regarding different values for $\mu$.

| Data | DCSI | Balance | $\mu$ | Noise |
|---|---|---|---|---|
| Adult (g) | 0.9867 | 0.4983 | 4 | - |
| | 0.9867 | 0.4983 | 5 | - |
| | 0.0880 | 0.8142 | 10 | 0.0005 |
| | **0.0878** | **0.8343** | **11** | **0.0005** |
| | 0.0091 | 0.8630 | 15 | 0.0005 |
| Adult (r) | 0.0000 | 0.4990 | 4 | - |
| | 0.0989 | 0.8480 | 5 | 0.0005 |
| | 0.0677 | 0.8499 | 10 | 0.0005 |
| | **0.0599** | **0.8603** | **11** | **0.0005** |
| | 0.0436 | 0.8633 | 15 | 0.0005 |
| Bank | 0.0706 | 0.9603 | 4 | - |
| | 0.0910 | 0.9649 | 5 | - |
| | **0.1357** | **0.9795** | **7** | **-** |
| | 0.1613 | 0.9942 | 10 | - |
| | 0.0567 | 0.9887 | 15 | - |
| Communities | 0.2028 | 0.8533 | 4 | 0.0005 |
| | 0.1325 | 0.8408 | 5 | 0.0005 |
| | 0.0963 | 0.8536 | 10 | 0.0005 |
| | 0.1461 | 0.8620 | 15 | 0.0005 |
| | **0.1533** | **0.9199** | **135** | **0.0005** |
| Diabetes | 0.0744 | 0.9606 | 4 | 0.0002 |
| | - | - | 5 | - |
| | - | - | 8 | - |
| | 0.1193 | 0.9691 | 10 | 0.0002 |
| | **0.0756** | **0.9622** | **15** | **0.0002** |

## A.3 RUNTIME EXPERIMENTS

In this section, we investigate FairDen's runtime in comparison to our competitors.

**Experiment design** We perform three experiments regarding the runtime of FairDen. We examine three scenarios: (a) increasing the **number of data points** (b) increasing the **number of numerical features** (dimension of the dataset), and (c) increasing the **number of ground truth clusters**. We generate datasets with the data generator DENSIRED (DENSIty-based Reproducible Experimental Data) (Jahn et al., 2024) for these experiments. The parameters used for the generation of the datasets are shown in Table 5. Note that the data generator generates entirely new datasets for each configuration. Consequently, not only do the parameters we intend to adjust vary, but the underlying structures do as well. We randomly assign a binary, sensitive attribute (by permuting an array composed of 50% ones and 50% zeros). The random assignment entails that we cannot guarantee the distribution within the clusters. As a result, the different datasets may not be directly comparable. However, this approach ensures equal conditions for all of the algorithms.

**Experiment setup** We perform five runs for each generated dataset. The runtime experiments are performed on a workstation with an AMD Ryzen Threadripper PRO 3955W, 250 GB RAM, and an RTX 3090. We set the cutoff time for each individual run to two hours.

Table 5: DENSIRED parameters for runtime experiments.

| Scenario | Number of data points | Dimensionality | Number of clusters |
|----------|----------------------|----------------|--------------------|
| (a) | 100, 200, 500, 1000, 2,000, 5,000, 10,000 | 10 | 2 |
| (b) | 2,000 | 5, 10, 50, 100, 1,000 | 2 |
| (c) | 2,000 | 10 | 2,3,4,5,6,7,8,9 |

**Results**  Figures 5-7 show runtime in seconds for the previously defined scenarios.

In the first scenario, we **increase the number of data points** in the dataset. The results are shown in Figure 5. We observe that our runtime increase is similar to normalized FairSC, with FairDen demonstrating slightly faster performance across the five runs. Both FairSC and Scalable Fair Clustering are faster. Fairlets, in contrast, has a significantly higher runtime that exceeds our cutoff of 120 minutes for datasets with more than 2000 data points.

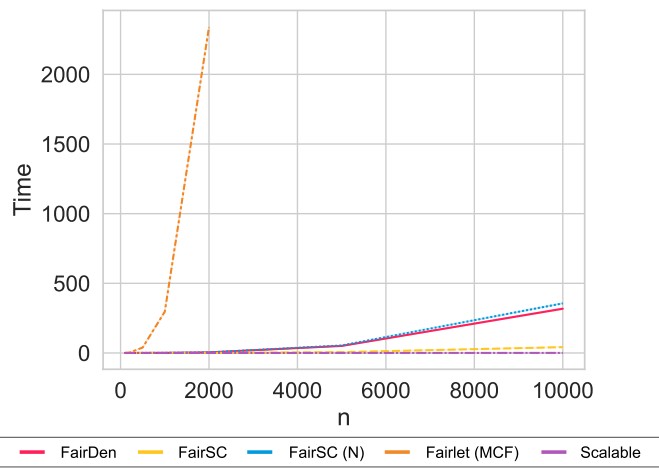

Figure 5: Runtime experiments for increasing numbers of data points $n$, with $n$ ranging from 100 to 10,000. Fairlet exceeds the time cutoff for more than 2,000 data points.

In the second scenario, we **increase the dimensionality** of the dataset. Since our proposed selection for $\mu$ depends on the dimensionality we regard two versions of FairDen in Figure 6: The bright pink color shows FairDen as used throughout the paper with the automatically computed $\mu$, the dark green denotes FairDen with a fixed $\mu$=5. The runtimes of both versions are very similar, ranging in a difference of around 0.1 seconds.

After introducing the comparison of the two FairDen versions we want to include a comparison across different methods, see the top of Figure 7. The left plot illustrates that the Fairlet approach does not yield comparable runtimes. We include a second version of the plot excluding Fairlets to show the differences between the remaining approaches better. Although all of the algorithms can calculate the result in less than 18 seconds we observe some variations in runtime. While FairSC and Scalable do not even need 2.5 seconds, normalized FairSC needs around 5. FairDen requires more time; the difference observed here between the various dimensions of the dataset is partly due to the fact that calculating the dc-distance takes a little longer. The calculations after determining the dc-distance are independent of the dimensions of the dataset.

In the third scenario, we **increase the number of ground truth clusters** in the dataset. The runtime results for this experiment are illustrated in the bottom of Figure 7. Note that the distribution of sensitive attributes or the underlying clustering structures or noise points, might vary heavily between the different datasets. We observe that especially in the bottom right part of Figure 7 for $k = 5$,

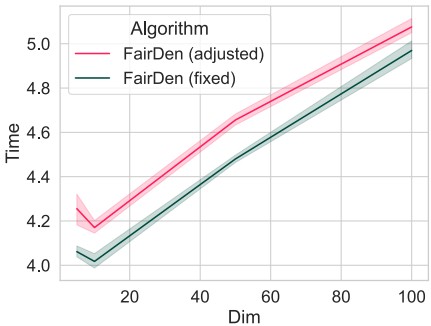

Figure 6: Runtime in seconds for FairDen with a fixed $\mu$=5 and $\mu$ automatically computed based on the dimensionality of the data as used throughout the paper.

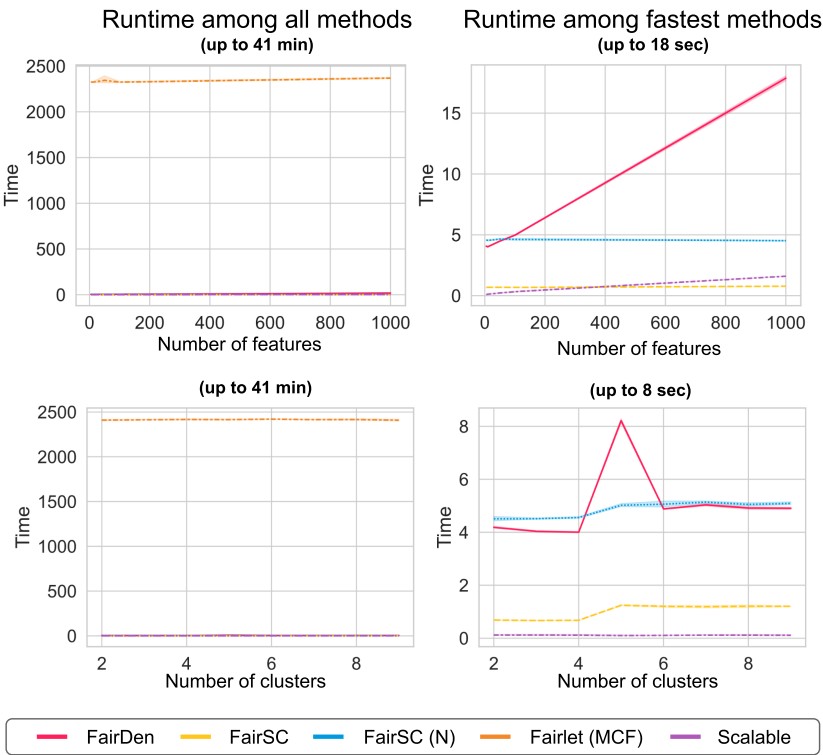

Figure 7: Runtime in seconds for experiments regarding (top) increasing numbers of features and (bottom) increasing numbers of clusters ($k$), for $k$ ranging from 2 to 9. Left: includes all of the algorithms. Right: algorithms excluding the slowest method, Fairlet MCF. For the experiments with increasing number of features, we used FairDen with a fixed value of $\mu$=5.

where FairDen has a peak in the runtime. Note that also the curves of FairSC and FairSC(N) have small bumps here. This effect might be due to an unfortunate positioning of noise points in this specific dataset. Except for this outlier, FairDen performs very similar to normalized FairSC. While FairSC and especially Scalable are significantly faster. The Fairlet approach needs again more time than any of the competitors with around 40 minutes per run.

## B  TECHNICAL BACKGROUND

The following section includes technical details regarding the dc-distance and DBSCAN.

### B.1  DBSCAN

DBSCAN (Density-Based Spatial Clustering of Applications with Noise) (Ester et al., 1996) is one of the most prominent clustering algorithms regarding the notion of density-based clustering. The algorithm differentiates between core points, border points, and noise points. The process of DBSCAN clustering is illustrated for $minPts$=5 in Fig. 8. The $minPts$ parameter denotes the minimum number of points required within a point's $\varepsilon$-radius for it to be regarded as a core point. Points with less than $minPts$ neighbors in their $\varepsilon$-neighborhood are considered border points if and only if they have at least one core point in their $\varepsilon$-neighborhood, otherwise the point is considered a noise point. DBSCAN implementations label noise points with -1.

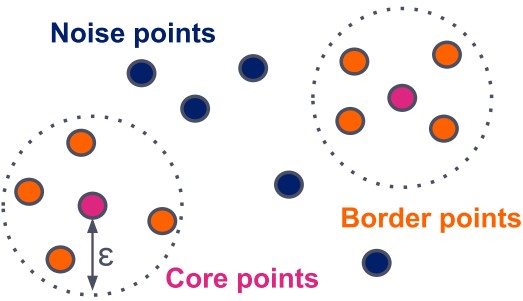

Figure 8: Illustration of DBSCAN clustering with $minPts$=5.

### B.2  DENSITY-CONNECTIVITY DISTANCE

The density-connectivity distance (dc-distance) is defined in Beer et al. (2023). It is based on the mutual reachability distance $d_m(x, y) = \max\left(d_{core}(x), d_{core}(y), d_{eucl}(x, y)\right)$, which is known from works as, e.g., DBSCAN (Ester et al., 1996), OPTICS (Ankerst et al., 1999), or HDBSCAN (Campello et al., 2013). The dc-distance represents the minimax (path) distance on the graph given by the mutual reachability distance $d_m$, see Eq. 16:

$$d_{dc}(x, y) = \min_{P \in \mathcal{P}} \max_{e \in p(x,y)} |e|, \tag{16}$$

where the length $|e|$ of an edge $e$ on a path $p(x, y)$ that connects points $x$ and $y$ is given by the dc-distance between the nodes that are connected by $e$. Intuitively, the dc-distance provides the smallest $\varepsilon$ so that two points are density-connected. As the dc-distance is based on the minimax distance, it inherits relevant properties that make it a well-defined (ultra-)metric (also known as rooted tree metric (Beer et al., 2023), which allows the representation of its distance matrix as a tree or hierarchy). The hierarchy established by the dc-distance works analogously to how the single link distance defines the traditional clustering hierarchy given by agglomerative single-linkage clustering. For more details, we refer the reader to the original paper by Beer et al. (2023).

Table 6: Properties of Real-World datasets. Sensitive attributes $a$, number of sensitive groups $g(a)$ for attribute $a$, number of numerical features $d_n$, number of categorical features $d_c$, DBSCAN parameters $minPts$ and $\varepsilon$.

| Dataset | Sens. Attr. ($g(a)$) | $d_n$(+ $d_c$) | $minPts$ | $\varepsilon$ |
|---|---|---|---|---|
| *Adult* (Kohavi et al., 1996) | race (5) | 5 *(+2)* | 4 | 2.1 |
| | gender (2) | 5 *(+2)* | 9 | 0.15 |
| | marital status (7) | 5 *(+2)* | 4 | 1.2 |
| *Bank* (Moro et al., 2014) | marital (3) | 3 *(+2)* | 4 | 1.5 |
| *Communities* (Asuncion & Newman, 2007) | black (2) | 67 | 10 | 3.25 |
| *diabetes* (Strack et al., 2014) | gender (2) | 7 | 10 | 0.45 |

## C  EXPERIMENT DETAILS

In the following, we give all experimental details and parameter settings, the definition of the used evaluation measure DCSI, and details of the real-world benchmark datasets.

### C.1  EXPERIMENT SETTINGS

The experiments are performed on a MacBook Pro, with an M2 Pro, and 16 GB of RAM using Python 3.9. Where possible, we integrated the author implementations for our competitors and the code reproducing every experiment for every competitor is included in our git.

The parameters of Scalable Fair Clustering are set according to the implementation (Backurs et al., 2019) so that the maximum desirable balance is set, $p = 1$ and $q = \left\lceil \frac{|S_x|}{|S_y|} \right\rceil$, with $|S_x| \leq |S_y|$.

The parameters for DBSCAN, $minPts$, and $\varepsilon$, were determined with a hyperparameter optimization. The criterion for the optimization is the DCSI score with a constant setting of $minPts_{\text{DCSI}} = 5$. The parameter space for $minPts_{\text{DBSCAN}}$ comprises $\{4, 5, 10, 15, 2d - 1\}$, with $d$ being the dimension of the dataset, and for $\varepsilon \in \{0.01, 0.05, 0.1, .., 0.5, 0.6, .., 2.5, 2.6, 2.8, 3, 3.25, 3.5, 3.75\}$. The final parameter settings are given in Table 6.

### C.2  DCSI

DCSI (Gauss et al., 2024) is an internal clustering validity index, that assesses the quality of a clustering regarding the connectedness and separability of the assigned clusters. The score relies on the notion of core points defined similarly to DBSCAN. The definition differs in terms of the parameters $minPts$ and $\varepsilon$, instead of defining them globally for all clusters, the approach of DCSI regards them as parameters that should be set individually for each cluster. The score includes only core points in its calculation. Core points are defined as points $x$ with at least $minPts_i$ data points from the same cluster $i$ having a distance $d(x, x') \leq \varepsilon_i$, note that $\varepsilon_i$ and $minPts_i$ are defined separately for each cluster $i$. The distance employed here is the mutual reachability distance.

$$DCSI(C) = \frac{2}{K(K-1)} \sum_{i=1}^{K-1} \sum_{j=i+1}^{K} DCSI(C_i, C_j) \qquad (17)$$

$$DCSI(C_i, C_j) = \frac{q}{1+q}, \text{where } q = \frac{Sep_{DCSI}(C_i, C_j)}{Conn_{DCSI}(C_i, C_j)} \qquad (18)$$

Separation is defined as the minimum distance between the core points of two clusters, $C_i$ and $C_j$, denoting how well the classes are separated. Connectedness is defined as the maximum distance between core points within one cluster. The choice of core points depends on $minPts$ and denotes the influence of noise/outliers. The proposed values are $minPts_i = 5$ for all clusters and $\varepsilon_i$ set to the median distance between all points x within the cluster and their $2 \cdot minPts_i$-*th* nearest neighbor in cluster $i$. The score, Eq. 17, is an accumulation of DCSI scores calculated for each pair of clusters

(cf. Eq. 18). Higher scoring denotes better clustering. Note that the score drops significantly when a certain number of points is assigned to another cluster than the optimal density-based clustering.

## C.3 REAL WORLD DATA

This section includes an overview of the included real-world datasets and our corresponding preprocessing.

*a) Adult.* The Adult dataset (Kohavi et al., 1996) includes 15 demographic features and categorizes 48,842 people based on their annual income (above or below 50,000 US-dollars). The sensitive attributes are *gender*, *race* and *marital status*. Depending on the setting, it has five numerical and up to two categorical features. Note that the distribution of groups within the individual sensitive attributes can vary largely, e.g., more than 70% of the data points belong to one protected *race*-group. We sampled 2000 data points from the dataset and removed duplicate entries based on the remaining features.

*b) Bank.* The bank marketing dataset (Moro et al., 2014) includes seventeen features that have been collected during marketing campaigns in Portugal between 2008 and 2013. The sensitive attribute *marital* includes three sensitive groups *married*, *divorced*, and *single*. The dataset includes a binary categorization whether a person subscribed to a term deposit or not. We include three numerical and two categorical variables. We sample the dataset to 5000 data points and remove duplicate entries based on the remaining features.

*c) Communities and Crime* Communities and Crime (Asuncion & Newman, 2007) includes data from the 1990 US census, law enforcement data from the 1990 LEMAS survey and crime data from the 1995 FBI's Uniform Crime Reporting (UCR). We use sensitive attributes as described in Kamiran et al. (2013) and Kamishima et al. (2012), yielding 67 numerical features. We exclude duplicate data points.

*d) Diabetes* The diabetes dataset (Strack et al., 2014) includes medical records regarding diabetes from 130 US hospitals. It is labeled according to whether a patient is readmitted within 30 days. We include seven numerical features and sample the number of data points to 5000, removing duplicates. The sensitive attribute is *gender*, divided into *female* and *male*.

