# OpenReview forum: "FairDen: Fair Density-Based Clustering"
_ICLR.cc/2025/Conference — ICLR 2025 Poster_

### Official Review · Reviewer_xP7u · 2024-11-02

**Soundness:** 3
**Presentation:** 3
**Contribution:** 3
**Rating:** 8
**Confidence:** 3

**Summary:**

This paper proposes a density clustering method based on fair learning. The authors claim that it is the first fair clustering algorithm capable of handling mixed-type data and multiple sensitive attributes simultaneously. The algorithm also includes the ability to detect noise.

**Strengths:**

1. The proposed method successfully detects fair clusters of arbitrary shapes while achieving balance.
2. The proposed density-based fair clustering method sounds interesting, and research in the field of fair machine learning also enhances contributions to the community.

**Weaknesses:**

1. The proposed method involves two existing ways to compute similarity matrices, and the form of the objective function is fundamentally consistent with spectral clustering. Although the authors emphasize that the distinction between their method and spectral clustering lies in the Laplacian matrix, this level of innovation is insufficient for ICLR.

2. The discussion on existing fair clustering methods is insufficient and should include the following relevant literature:

[1] Chen X, Fain B, Lyu L, et al. Proportionally fair clustering[C]//International conference on machine learning. PMLR, 2019: 1032-1041.
[2] Esmaeili S, Brubach B, Tsepenekas L, et al. Probabilistic fair clustering[J]. Advances in Neural Information Processing Systems, 2020, 33: 12743-12755.
[3] Esmaeili S, Brubach B, Srinivasan A, et al. Fair clustering under a bounded cost[J]. Advances in Neural Information Processing Systems, 2021, 34: 14345-14357.
[4]Dickerson J, Esmaeili S, Morgenstern J H, et al. Doubly constrained fair clustering[J]. Advances in Neural Information Processing Systems, 2024, 36.

3. There are several omissions in the paper. For instance, vol(c) should be vol(c_l),  and the output of Algorithm 1 is also unclear.
4. The latest comparison method is from five years ago, which raises questions about the algorithm's effectiveness. Even without involving the most recent fair clustering methods, the referenced conventional clustering methods should not be limited to only DBSCAN.

**Questions:**

1. Why does Table 3 not include results from other fair comparison methods?
2. Do the experimental results depend on the parameters? The authors did not conduct experiments on this aspect.
3. How does the proposed method address the issue of sensitive attribute combinations (e.g., Black women), particularly regarding the physical meaning of Equation 7? Could the authors provide a detailed explanation?

---

> ### Author Response · Authors · 2024-11-20
> **Part 1: Answer to weak points**
>
> Dear Reviewer,
>
> thank you for your insightful feedback and time!
>
> * W1: We would like to emphasize that fairness in density-based clustering has not been explored before. One of the key innovations in our work is the observation that dc-distance allows us to incorporate fairness into density-based clustering by using spectral clustering, though this is just one of the novel elements of our approach. In particular, FairDen is designed to handle non-binary sensitive groups, multiple sensitive attributes, and categorical data features— ignored by existing fair methods, despite their relevance in real-world scenarios. We are pleased that, despite the simplicity and clarity of our method, it effectively addresses several important issues and introduces unique contributions that differentiate it from competing approaches (see Table 1 for a summary).
>
>   1. To the best of our knowledge, no other fair clustering method inherently includes handling categorical attributes, even though it is very typical for data with sensitive attributes (which are categorical) to also include other categorical attributes. While we include the similarity measure based on Goodall1 (Boriah et al., 2008), this approach cannot be used for center-based approaches.
>   2. Multiple and non-binary sensitive attributes: Many fair methods, do not work for non-binary sensitive attributes, as it is a more complex problem – that, however, FairDen solves successfully.  While two of the works referenced by you ([2] and [3]) also take into account non-binary sensitive attributes, integrating multiple sensitive attributes still poses a difficult problem they do not achieve.
>
> * W2: We appreciate the suggestions for additional related work. In the interest of space, we originally limited the discussion to the core scope of the paper: group-level fair clustering optimizing the balance of sensitive attributes in combination with arbitrarily shaped clusters. Following your suggestion, in the revised version, lines 479-503, we added more details to clarify further this emerging research into a broader context to help to delineate our work more clearly. However, note that the suggested methods, in contrast to FairDen, do not aim at density-based clusters. Additionally, they solve different objectives than FairDen, thus, we did not include them in the comparative experiments: [1] is based on proportional fairness and not on balance, [2] assumes the sensitive group (color) is assigned probabilistically rather than deterministically, and [3] optimizes fairness as an objective function up to a determined upper bound of clustering cost. As [4] also includes group-level fairness, we are interested in further investigating it for its usage as a post-processing step for achieving fairness in density-based clustering - however, this is future work and out of the scope of this paper.
>
> * W3: Thank you for the note, we fixed this typo in the revision and added some more details on the exact shape of the output of Algorithm 1.
>
> * W4: As there are no fair density-based clustering algorithms besides FairDen yet, we compare FairDen to other group-level fair algorithms that can potentially find arbitrary-shaped clusters, as they bear some relationship to density-based clusters. We furthermore included the conventional clustering DBSCAN only as a baseline of which FairDen is the fair version, similar to related work (e.g., Kleindessner et al.) where fair spectral clustering is compared to its vanilla version (conventional spectral clustering). Since our focus is on density-based clustering we do not include centroid-based conventional clustering methods.

---

> > ### Comment · Reviewer_xP7u · 2024-11-26
> >
> > The rebuttal addresses my concerns. I would like to raise the score.

---

> > > ### Author Response · Authors · 2024-11-26
> > >
> > > Dear Reviewer,
> > >
> > > Thank you very much for your valuable feedback and recognizing our efforts to improve the paper!

---

> ### Author Response · Authors · 2024-11-20
> **Part 2: Answer to questions**
>
> In the following, we answer the questions (Q1-Q3):
>
> * Q1: This experiment assesses the inclusion of categorical data into FairDen vs excluding categorical data. As our competitors cannot handle categorical data, and also the evaluation measure DCSI cannot be applied to categorical attributes, we show the results of the datasets without the categorical attributes in Table 2. Thus, Table 3 contains only the additional results from methods that are capable of handling categorical attributes, i.e., DBSCAN and FairDen.
>
> * Q2: While we automatically computed our only hyperparameter $\mu$ based on the data for all experiments, we had tested its influence on the clustering results. Because of space limitations, the results were included in the paper only in the appendix (A.2), but we refer to them more prominently in the revised version (l.302).
>
> * Q3: A sensitive group can also be a combination like black women. Thus, for example, in Equation 7 there are two ways to compute the balance of a cluster. First, one could be interested in the balance regarding the individual sensitive attributes, i.e., "is the cluster balanced regarding gender?" And "is the cluster balanced regarding race?" In this case, both balance values are handled separately. The other option, which also corresponds to the solutions FairDen is finding, is to regard both sensitive attributes at once, i.e., combining them. Thus, for $a$ many sensitive groups for a sensitive attribute $A$ and $b$ many sensitive groups for another sensitive attribute $B$, we have up to $a$ times $b$ many combined sensitive groups (or fewer, as not every value combination might occur in the data).

---

### Official Review · Reviewer_DheD · 2024-11-02

**Soundness:** 3
**Presentation:** 3
**Contribution:** 3
**Rating:** 8
**Confidence:** 4

**Summary:**

The paper presents FairDen, a density based clustering algorithm with group fairness considerations. As it is written in the paper, it is a novel approach, forming fair clusters that supports mixed-type data types, and multiple sensitive attributes, with a weighting mechanism. The contributions include the formulation of the algorithm in an analytical way, and a standard set of experimentation.

**Strengths:**

Overall, this is a solid work, but to name a few strengths, I can mention these:
1- The method is clearly described with some sound step by step derivation of the last optimization objective.
2- The implementation is submitted as well.
3- The experiments section cover a good set of algorithms and datasets.

**Weaknesses:**

There are a few things that could be better about this paper. Given these are addressed, I am willing to increase my scores:

1- The limitations, such as the reduced performance on very small sensitive minority groups, are scattered throughout the text. A dedicated limitations section before the conclusion that consolidates these points would be beneficial.
2- Figure 1 is a canonical figure, but could be more clear. Mainly because it is too visually dense and the description of it is broken down into pieces being in introduction section in the beginning and the related works towards the end. As an option, breaking it up into multiple figures with more information rich descriptions, especially, on the third colour used for the DBSCAN sub-figure would be beneficial.
3- In the complexity analysis section, it is mentioned "Note that runtime can be reduced significantly, e.g., by using the GPU for matrix computations.". Although it is becoming a common knowledge that using parallel approaches can decrease practical runtime, this claim does not seem to be supported by an experiment. If there is explicit evidence to support it, it should be added. Otherwise, removal is advised.
4- There are some writing issues that could be fixed:
 4.1- Setting the most common abbreviations such as i.e., and e.g. aside, the paper has numerous abbreviations that is best to be omitted.
 Here are the list I found: "iff" on line 92, s.t. on lines 117, 221,523,... w.r.t on lines 416, 418 ...
 A full review of the paper and spelling out the non-standard abbreviations is advised.
 4.2- The line numbers of Algorithm 1 are not appropriately referenced in the text. For example: on line 201, it is ambiguous that (line 4) is
 mentioning Algorithm 1 in the appendix. Making the reference explicit make the paper more readable.

Here is a suggestions rather than stating weaknesses:
The authors did a great job with "Multiple sensitive attributes"and the following figure 3. It could be great to have a general interpretation section on the clusters that this algorithm forms.

**Questions:**

Is there minimum number on the count of data types for the algorithm to be effective? In another words, how many numerical features are needed, relatively to be able to form fair clusters given various number of sensitive attributes? Has there been any more studies done on the algorithm that could be included to give more insights on this aspect of the algorithm?

---

> ### Author Response · Authors · 2024-11-20
>
> Dear Reviewer,
>
> Thank you for your time and the helpful feedback!
> We address each of the weaknesses and questions individually. We highlight changes in the revised pdf version in blue color.
>
> * W1: Thank you for the suggestion of adding a limitations section! We added the section in the revised version, see lines 288-292.
>
> * W2: We have simplified Figure 1 based on your comment, and enhanced it with a legend, along with an updated caption to improve clarity.
>
> * W3: We believe that GPU parallelization is a promising direction for future work, especially considering the complexity of the problem that other reviewers also noted. Thus, we would like to keep mentioning this idea in the paper, despite no experimental backup. Thus, in the revised version, we clarified it is future work and accordingly moved it to the conclusion (line 540) – is this an appropriate solution to your concern?
>
> * W4: Thank you for the detailed remarks, we directly followed them and improved the paper accordingly.
>
> * Regarding your last paragraph with the suggestion: Thank you! We added a paragraph regarding the interpretation of clusters in the analysis section, lines 274-281.
>
> * Q1: This is a very interesting point of discussion, the number of numerical features or sensitive attributes do not play as important a role as the distributions they represent. In the extreme case, it might be that there is only one sensitive attribute with feature values distributed such that fairness is in direct contradiction to density-based clustering using numerical features. On the other extreme, even for few numerical features and many sensitive attributes, if the values in sensitive attributes are evenly distributed among the numerical feature clusters, the problem is solved effortlessly. Still, we would expect that fewer sensitive attributes, more numerical features, as well as a large size of the data set, might be favorable to fair density-based clustering. In sum, we consider it a highly data-dependent question, which could be interesting for case-based studies in fairness applications.

---

> > ### Comment · Reviewer_DheD · 2024-11-22
> >
> > Thank you for your revision and addressing my concerns. I believe these changes have strengthened the paper making it a good contribution to the field. So, I am happy to increase my rating.

---

> > > ### Author Response · Authors · 2024-11-26
> > >
> > > Dear Reviewer,
> > >
> > > Thank you very much for your valuable feedback and recognizing our efforts to improve the paper!

---

### Official Review · Reviewer_Jc4S · 2024-11-02

**Soundness:** 3
**Presentation:** 2
**Contribution:** 2
**Rating:** 5
**Confidence:** 4

**Summary:**

The paper introduces a new density-based clustering method called FairDen - it is designed to incorporate fairness without compromising the density-based structure of the data.  Fairness has been studied in clusterings, but density-based clustering algorithms have not been studied through the fairness lens.

Density-based clustering methods come with their own merits - eg, they can handle categorical data (or in general, mixed data types) and can handle noise.  FariDen inherits these nice properties.  In addition, it can handle mutiple sensitive attributes simultaneously.  Experiments on real-world datasets show that FairDen produces fairer and more balanced clusters compared to other density-based algorithms and better captures density-based clusters than existing fairness-oriented algorithms - thus achieving the better of both worlds.

Technically, the key idea is based on Eqn (11) - how to incorporate the fairness constraint into the normalized cut objective.  Eqn (12) essentially summarizes the paper's contribution.  The task then is show how to solve (12) via spectral clustering.  Luckily, the fairness constraint is (rigid and) easy enough to be able to reshape the constrained problem into an unconstrained spectral clustering problem.

**Strengths:**

* DBScan is widely used and practical clustering algorithm and a fair version of it is a natural question

* The ability to handle categorical data, which is more of a property of DBScan.

* The proposed method can handle several sensitive attributes.

* The paper has good experimental results, showing good outcomes on both the balance aspect as well the density aspect

**Weaknesses:**

* The ideas are incremental and largely based on based on existing work.  The dc distance (continuous) formulation is from the KDD paper of Beer et al and solving such problems with spectral methods is quite standard, even in the fairness literature.

* The fairness constraint it treated in a very rigid way, which makes the problem way too simple.  (Eqn (12)).  A more relaxed accommodation of fairness (instead of equality, an approximation in some matrix-norm sense) would be more meaningful here.

* The scalability of the algorithm is also unclear for large datasets (complexity=O(n^3)), given its reliance on spectral methods.  (This criticism is applicable to some other but not all work on fair clustering).  Combinatorial or sketching algorithms might have better running times.

* The writing is extremely poor.  It assumes that the reader has a lot of background on the KDD 2023 paper and even about DBScan.  The paper is hardly self-contained - the authors could have written more clearly or provided missing details in the Appendix.  For example, they assume the reader knows what an ultrametric is or an ultrametric leads to a hierarchy ... there are many such instances.  See minor comments below.

Minor comments:

93: not sure e-range is standard -> use eps-radius?

89-93: sloppily written - hard to parse

116: the ddc distance is taken from Beer et al - not even defined in the text

119: what is the hierarchy you are talking about?

127-134: hard to parse

145: state I_k is the kxk identity matrix

161: The model itself is not clear here: what does single index S_i mean?  Is the setting as simple as a bipartite graph of X times S, where edges from X to S denote membership of a user in a sensitive group?  At the end of the sentence, i is used to index attributes where in the beginning, it is used to index the user (data). In fact, for S_{ij} what does i and j stand for?

162: what is k? (i assume # clusters)

167: r_ij is not the whole dataset (as written it reads that way)

176: it might be enough to index balance_{ij} instead of balance_{S_{ij}}

197: what is the quantifier for l in (10)

219: why does the inverse ultrametric even matter?

305: Aren't there are scenarios where heavy bias is inevitable and hence replicating the biases in the clusterng is also inevitable?

357: again, what is k?

**Questions:**

The constraint in (11) is very rigid.  Could you methods be adopted to relax it to say an approximation of it?

---

> ### Author Response · Authors · 2024-11-20
>
> Dear Reviewer,
>
> Thank you for your time and your detailed review, especially the minor comments for improving our paper are much appreciated! We follow many of your suggestions for the revision and highlight changes in the revised PDF version in blue color. In the following, we address your questions:
>
> * W1: We would like to point out that the integration of fairness into density-based clustering has not been studied before. Leveraging the dc-distance to incorporate fairness into density-based clustering is neither incremental nor our only contribution: Concretely, FairDen admits non-binary sensitive groups, multiple sensitive attributes, and categorical data aspects that other fair methods neglect despite their relevance for real-world data. We appreciate that our approach is still perceived as simple and coherent, even though it addresses several important aspects and novel contributions that our competitors do not (see Table 1 for an overview).
>
> * W2  / Q1:  Approximating fairness instead of optimizing it is an interesting idea for future work! As we note in the conclusion of the paper, also extending the approach to different notions of fairness, or investigation of related constrained problems are interesting open questions. However, in this paper, our goal is to optimize the fairness/balance.
>
> * W3: We conducted some runtime experiments that we include in the revised version of the pdf, to set the computational complexity into context. Thank you for bringing up combinatorial or sketching algorithms, we see this as a promising direction for future work. However, as FairDen is the first fair density-based approach, our focus was on the conceptual contribution and quality, but not yet on runtime optimization.
>
> * W4: We follow your suggestion and added more details in the revised version accordingly and added corresponding sections in the appendix (Appendix B “Technical Background“ ). Furthermore, we removed interesting, but not necessary parts (like the dc-distance being an ultrametric) from the paper and added details in the Appendix instead, see also our explanation regarding minor comment for l.119 at the end.
>
> Regarding the minor comments:
> Thank you for your very detailed feedback! We addressed most of them directly, for those with a question, we provide some comments:
>
> * “161: The model itself is not clear here: what does single index $S_i$ mean? Is the setting as simple as a bipartite graph of $X$ times $S$, where edges from $X$ to $S$ denote membership of a user in a sensitive group? At the end of the sentence, $i$ is used to index attributes where in the beginning, it is used to index the user (data). In fact, for $S_{ij}$ what does $i$ and $j$ stand for?”
>   - We had omitted the index $i$ in $x_i \in X$ for simplicity, as it is not required in the rest of the paper, but we can see now that it might indeed be confusing regarding its later use in combination with $S$.
>   - $S_i$ refers to any of the sensitive attributes (see line 162), where the number of different sensitive attributes is given by $a$, which we introduce earlier and more explicitly in the revised paper (than in the current line 179).
>   - Each sensitive attribute (e.g., gender, race, age, …) can have several, usually categorical values. E.g., for gender, “male”, “female”, “diverse”. Those define three sensitive groups with regard to the sensitive attribute. We refer to the sensitive attribute with the first index, $i$, and to each of the groups for this attribute, using a second index, $j$.
>   - One could model the membership of an object $x$ in a sensitive group $S_{ij}$ with an edge in a bipartite graph of $X \times |S_{ij}|$ . Every object $x$ has then exactly $a$ many edges (one for every sensitive attribute).
>
> * “305: Aren't there are scenarios where heavy bias is inevitable and hence replicating the biases in the clusterng is also inevitable?”
>   - Yes if the dataset only consists of data points that cannot be separated into fair clusters, i.e., the sensitive groups across different sensitive attributes are non-separable (intersection discrimination) there exists no fair solution. This observation is true for any fair clustering notion.
>
> * “119: what is the hierarchy you are talking about?“
>   - The hierarchy is defined by the dc-distance and works analogously to how the single link distance defines the standard clustering hierarchy given by agglomerative single-linkage clustering. We have tried to clarify this in the revised paper.

---

> > ### Comment · Reviewer_Jc4S · 2024-11-28
> > **Response**
> >
> > Thank you for your participation and the responses.

---

> > > ### Author Response · Authors · 2024-11-29
> > >
> > > Thank you for acknowledging our responses. Could you kindly provide us with an indication of whether you consider these changes sufficient for addressing your comments, in particular the presentation? If so, we would appreciate you considering to raise your scores accordingly.

---

> > > > ### Comment · Reviewer_Jc4S · 2024-12-03
> > > >
> > > > I am unconvinced about the novelty.  I will maintain my score.  Thanks.

---

### Official Review · Reviewer_ygXB · 2024-11-04

**Soundness:** 3
**Presentation:** 3
**Contribution:** 3
**Rating:** 6
**Confidence:** 3

**Summary:**

The paper proposes a method for fair density-based clustering, whereby the discovered groupings satisfy certain fairness criteria in the context of sensitive attributes.  Whilst fairness criteria have been explored  for other clustering methods, this appears to be the first work that addresses it for density based clustering.  The key insight of the paper is to model density connectivity between points using a continuous representation, that facilitates a so-called fairness-balancing procedure.  The proposed method is distinguished in its ability to handle aspects such as categorical attributes, noise and mixed data.

**Strengths:**

-fills a gap in the fair clustering literature through its focus on density based clustering.  Density based methods can be particularly attractive through their ability to handle scenarios with categorical data and noise

-proposed method is able to handle scenarios with categorical attributes and mixed type data, in contrast to some baselines.  It can also handle scenarios with multiple sensitive attributes.  It is thus quite “expressive” in terms of fair clustering approaches.

-its formulation leverages well known and principled approaches through spectral clustering

-solid experimental performance in terms of balance for a single sensitive attribute.  Some exploration of the benefits of multiple sensitive attributes.

**Weaknesses:**

-approach is potentially expensive computationally being O(d*n^3).  The experiments don’t explore runtime of the method, particularly for larger datasets.  The selected real world datasets appear to be of smaller size.
  Can this aspect be explored further to provide more insight?

-The proposed method allows inclusion of noise points, which is a strength.  However, it is not so clear from the experiments what can be the advantages of handling noise.  Indeed noise points are specifically excluded in some of the evaluations.  Could a specific experiment be designed to evaluate the role of noise for fair clustering?

**Questions:**

Figure 1 – it is hard to visually distinguish which methods are producing better clusters.  They all look rather similar.  It’s also hard to visually match the balance and dcsi values to the clusters.  DCSI is mentioned in the caption before it has been defined in the paper.

In Table 2 and Table 3, the ARI/NMI seem very low (close to zero).  Earlier it is mentioned that higher is preferable.  How is one supposed to interpret such low values – is it a drawback of the clustering method?

To what extent would be the approach for the fairness constraint described in 2.2, be extendible to different or multiple fairness constraints?   Does feasibility need to be worked through on a case by case basis?

Can scalability and noise be examined further, as described above.

---

> ### Author Response · Authors · 2024-11-20
>
> Dear Reviewer,
>
>
> We sincerely appreciate your feedback and time, thank you! We highlight changes in the revised pdf version in blue color.
> * Q1: We changed Figure 1 to be more clear and added a legend which shows that the clusters are given by the colors and the sensitive attributes are given by the shapes of the objects. Thus, continuous areas of the same color represent ‘good’, density-based clusters. We furthermore include a short description of DCSI and balance in the caption.
>
> * Q2: Both ARI and NMI evaluate the clustering result regarding the ground truth labels. However, since we aim at finding group-level fair density-based clusterings rather than clusterings reflecting the possibly unfair ground truth labels, the external evaluation metrics do not offer much insight into quality of fair clustering. We included them merely to provide some background information about the contrast between ground-truth labels, and also density-based clustering (DBSCAN), and fair clustering alternatives.
>
> * Q3: Thank you for this question, this is an interesting idea and a promising direction for future work. Some constraints like individual-level fairness could most probably be included, however, there would be a trade-off between the different fairness notions (and also, the clustering quality) if their optimal solutions are not equal. In the extreme case, the combination of constraints may not admit a solution. Indeed, the feasibility largely depends on the data distribution, in particular in regards to the sensitive attribute value combinations, and would require a case-based analysis for a given dataset.
>
> * Q4 + W1: We conducted some runtime experiments which we added to the appendix of the revision (Appendix A.3). FairDen had comparable runtimes to the normalized spectral-based competitor. Scalable Fair Clustering and Fair SC are faster. In contrast, the Fairlet-based approaches had significantly higher runtimes.
>
> * W2: Generally, the ability to detect noise is a favorable property of clustering algorithms, as it avoids degenerate clusters affected by noise - in fact, it is arguably one of the reasons for the popularity of DBSCAN. In our case, noise indicates individuals that should be investigated manually since they might differ from the population considerably. Note that including noise points as singleton clusters in the computation of the balance would cause the full clustering to have a balance score of 0, which is why we included them only indirectly for the respective experiments as described in Sections 3.1 and 3.2. Detecting noise points in a fairness setting improves the handling of intersectional fairness, especially for very small minorities, as it points to special cases, that may benefit from including the user in the loop.

---

> > ### Comment · Reviewer_ygXB · 2024-11-27
> >
> > Thanks for your responses.   I intend to maintain my score.

---

### Meta-Review · Area_Chair_47Qb · 2024-12-17

**Metareview:**

This paper introduces FairDen, a novel density-based clustering algorithm that incorporates fairness constraints.  The authors address a gap in the literature by extending fairness considerations to density-based clustering, which offers advantages in handling mixed data types and noise compared to traditional methods. FairDen achieves fairness by modeling density connectivity with a continuous representation and integrating a fairness-balancing procedure into the clustering objective. Experiments demonstrate that FairDen produces fairer clusters than existing density-based methods while effectively capturing density-based structures.

Reviewers acknowledge the practical importance of developing a fair version of DBSCAN, a widely used density-based clustering algorithm. They appreciate FairDen's ability to handle scenarios with categorical attributes and mixed-type data, which is a significant advantage over existing fair clustering methods. The experimental results are viewed positively, supporting the effectiveness of the proposed approach.

However, reviewers also raise some concerns:

- Limited Novelty: While FairDen addresses a relevant problem, some reviewers find the technical novelty to be somewhat incremental, building upon existing fairness and density-based clustering techniques.
- Computational Complexity: The proposed method has a high computational complexity, potentially limiting its scalability to large datasets.
- Writing Quality: The clarity and presentation of the paper could be improved.
- Related Work: The discussion of related work could be more comprehensive, providing a deeper analysis of existing fair clustering methods and their limitations.

Recommendation:

Despite the limitations, the paper makes a valuable contribution by introducing fairness to density-based clustering and addressing practical challenges related to mixed data types.  I recommend accepting this paper as a poster.

**Additional Comments On Reviewer Discussion:**

The discussion where nice and help reviewers solve few questions on the paper

---

### Decision · Program_Chairs · 2025-01-22

Accept (Poster)